# PARP1: Liaison of Chromatin Remodeling and Transcription

**DOI:** 10.3390/cancers14174162

**Published:** 2022-08-27

**Authors:** Wen Zong, Yamin Gong, Wenli Sun, Tangliang Li, Zhao-Qi Wang

**Affiliations:** 1State Key Laboratory of Microbial Technology, Shandong University, Qingdao 266237, China; 2Leibniz Institute on Aging—Fritz Lipmann Institute (FLI), 07745 Jena, Germany; 3College of Basic Medical Sciences, Shenzhen University Medical School, Shenzhen 518055, China; 4Faculty of Biological Sciences, Friedrich-Schiller-University of Jena, 07743 Jena, Germany

**Keywords:** PARP1, PARylation, chromatin, transcription, inflammatory response

## Abstract

**Simple Summary:**

Poly(ADP-ribose) polymerase 1 (PARP1) is perhaps the most studied member of the PARP superfamily and participates in numerous cellular processes. PARP1 inhibitors have been approved as drugs to treat various cancers in clinics, based on its role in DNA repair. Yet, there is a growing body of evidence showing multitasking function of PARP1 in regulation of gene expression. In this review article, we discuss the current knowledge of PARP1 and its conducted enzymatic process, i.e., PARylation, with an emphasis on gene expression by the interaction with transcription factors and regulation of chromatin conformation, dependent or independent of DNA damage. The molecular action mode of PARP1 in gene transcription may present as a potential target for therapeutic intervention of inflammation-related diseases and also for cancer therapy.

**Abstract:**

Poly(ADP-ribosyl)ation (PARylation) is a covalent post-translational modification and plays a key role in the immediate response of cells to stress signals. Poly(ADP-ribose) polymerase 1 (PARP1), the founding member of the PARP superfamily, synthesizes long and branched polymers of ADP-ribose (PAR) onto acceptor proteins, thereby modulating their function and their local surrounding. PARP1 is the most prominent of the PARPs and is responsible for the production of about 90% of PAR in the cell. Therefore, PARP1 and PARylation play a pleotropic role in a wide range of cellular processes, such as DNA repair and genomic stability, cell death, chromatin remodeling, inflammatory response and gene transcription. PARP1 has DNA-binding and catalytic activities that are important for DNA repair, yet also modulate chromatin conformation and gene transcription, which can be independent of DNA damage response. PARP1 and PARylation homeostasis have also been implicated in multiple diseases, including inflammation, stroke, diabetes and cancer. Studies of the molecular action and biological function of PARP1 and PARylation provide a basis for the development of pharmaceutic strategies for clinical applications. This review focuses primarily on the role of PARP1 in the regulation of chromatin remodeling and transcriptional activation.

## 1. Introduction

Poly(ADP-ribosyl)ation (or PARylation) is an abundant post-translational modification (PTM) that regulates a variety of cellular pathways in both prokaryotes and eukaryotes. The poly(ADP-ribose) polymerases (PARPs) are a major family of enzymes capable of modifying proteins by PARylation [1]. PARPs can be found in almost all subcellular compartments within the cell, including in the nucleus, cytosol, mitochondria, endoplasmic reticulum, etc. They use intracellular nicotinamide adenine dinucleotide (NAD^+^) to modify acceptor proteins with mono-ADP-ribose (MAR) moieties or long (up to 200 units) branched poly-ADP-ribose (PAR) [1]. PARP family contains 17 members in humans, but not all of them are enzymatically active. PARP1, PARP2 and tankyrases (i.e., PARP5a (tankyrase-1) and PARP5b (tankyrase-2)) are bona-fide PARPs, generating PARs [1,2]. PARP13 is a pseudo-enzyme without catalytic activity, although it possesses a structurally similar domain to the catalytic domain of active PARP members [1,3]. The activity of PARP9 has not been fully clarified [1,2,3]. Others are mono-ADP-ribose transferases that generate MAR [1,2]. PAR levels are tightly controlled in the cell by timely degradation via ADP-ribosyl hydrolases, including PAR glycohydrolase (PARG), ADP-ribosyl hydrolase 3 (ARH3), macrodomain-containing proteins MacroD1 (LPR16) and MacroD2, as well as terminal ADP-ribose hydrolase (TARG1 or C6orf130). While PARG and ARH3 degrade the PAR chains, MacroD1, MacroD2 and TARG1 remove the protein proximal mono-ADP-ribose and enable the complete reversal of the modification [4].

PARP1 (also known as diphtheria toxin-like ADP-ribosyltransferase, ARTD1), is the founding member of the PARP superfamily. It is the most active member of the PARPs, and believed to execute approximately 90% of total PARylation activity [5,6]. PARP1 adds a negatively charged PAR to various acceptor proteins, including DNA repair proteins, histones, chromatin remodelers, transcription factors and signal transduction elements, yet mainly to PARP1 itself (called auto-PARylation). PARP1 has been widely and extensively studied. The most known function of PARP1 is its function in DNA damage repair, including pathways of base excision repair (BER), the repair of single-strand breaks (SSBs), double-strand breaks (DSBs, by homologous recombination (HR), non-homologous end joining (NHEJ) and alternative end-joining (Alt-EJ)), as well as of stalled replication forks [7,8]. In addition, PARP1 has been shown to be activated by chromatin conformation changes, and thereby modulates chromatin remodeling, which can be independent of DNA breaks [9,10]. Another well-known DNA break-independent function of PARP1 is its role in transcriptional regulation. PARP1 has been involved in various aspects of the transcription process through a variety of mechanisms, including regulation of chromatin remodeling, DNA methylation and coregulation of transcription factors. There have been comprehensive reviews regarding the function of PARP1 and PARylation in DNA repair and damage signaling [7,8]; this review article focuses on the function of PARP1 in chromatin remodeling (chromatin conformation change, but DNA break independent), DNA methylation and transcription (as a cofactor for transcription factors and also modulator of chromatin).

## 2. PARP1 Structure and Activation

The *PARP1* gene, the first PARP family member, was cloned in 1987 [11,12,13] and mapped to chromosome 1 (1q42.12) in humans. Its encoded protein PARP1 contains 1014 amino acids, with a molecular weight of approximately 113 kDa [14]. PARP1 is an evolutionarily conserved, multifunctional enzyme that is found in all eukaryotes, except yeast, enigmatically [15]. Structurally, PARP1 consists of the following three major functional domains (Figure 1A): an *N*-terminal DNA-binding domain (DBD), an automodification (or auto-PARylation) domain, and a catalytic domain in the C-terminus (CAT) that contains a highly conserved sequence within the active site, defined as the PARP signature.

The *N*-terminal domain contains three zinc finger DNA-binding domains, namely ZFI, ZFII and ZFIII. ZFI and ZFII are critical for the recognition of various DNA structures with high affinity for DNA [16,17,18]. ZFIII is used to couple DNA-binding and catalytic activities of PARP1 and has a structure and function that different from ZFI and ZFII [18,19]. In the N-terminus, there is a nuclear localization sequence (NLS) (KRK-X (11)-KKKSKK) that leads PARP1 to the nucleus. During apoptosis, PARP1 is cleaved by caspases at the conserved site _211_DEVD_214_, generating 24 kDa and 89 kDa fragments. The auto-modification domain has a BRCA1 C-terminal (BRCT) domain and tryptophan-glycine-arginine-rich (WGR) domain. The BRCT domain provides sites of auto-PARylation and regulates protein–protein interaction. A recent study proposed that this BRCT domain can bind to intact DNA without activation of PARP1 and mediate rapid movement of the PARP1 molecule to scan damaged DNA [20]. The WGR domain, ZFI and ZFIII together interact with DNA to link the DNA damage interface to the catalytic domain [18]. Besides the BRCT domain, the auto-modification domain is rich in serine, glutamate and lysine residues, which are the main sites of auto-PARylation. Serine is a major residue for ADP-ribosylation upon DNA damage [21,22]. The C-terminal catalytic domain contains the (ADP-ribosyl) transferase (ART) motif, called the PARP signature, which is a NAD^+^ binding site and the residues that contribute to the initiation, elongation, and branching of PAR [23].

PARP1 can be activated by multiple stimuli, including but not limited to, DNA damage, ERK1/2 [24], hormones [25], JIL-1 [26] and NAD^+^ [27]. The activated PARP1 forms long and branched PAR chains mainly to its auto-modification domain but also to other acceptor proteins, such as histones, chromatin regulators and transcription factors. PARP1 uses NAD^+^ as a substrate for PAR formation, while releasing nicotinamide. PAR chains are then rapidly catabolized by PAR degrading enzymes, including PARG, ARH3, MacroD1, MacroD2 and TARG1 [28,29,30,31]. PARG is the main and robust enzyme that cleaves the ribose–ribose bonds of PAR to release free PAR and ADP-ribose (ADPr) [32]. ARH3 can also catalyze the removal of PAR but not MAR [29]. Furthermore, ARH3 removes the O-acetyl group from the NAD^+^ metabolite O-acetyl-ADP-ribose [33]. TARG1 cleaves the terminal ADP-ribose moiety, resulting in the release of the last protein proximal ADPr moiety [34]. Finally, the free PAR can be recycled for ATP production and PARP1 can be activated again [35] (Figure 1B).

## 3. PARP1 and Chromatin Remodeling

Chromatin remodeling, a dynamic modification of chromatin architecture from a condensed state to a DNA-binding protein accessible state, plays a crucial role in regulating various cellular processes [36]. Chromatin consists of genomic DNA and proteins. The major proteins in chromatin are histones, which compose of linker histones (H1) and core histones (H2A, H2B, H3 and H4) and help package DNA into a compact form that fits into the cell nucleus. Chromatin restricts the accessibility of DNA-binding factors to DNA [37]. Cells have evolved the mechanism of chromatin remodeling to change chromatin conformation that allows for better or full access of DNA repair machineries, DNA replication factors, transcription factors and chromatin condensation factors to DNA [38]. PARP1 is an important factor that participates in this process, and hence regulates numerous cellular processes, including DNA repair, DNA replication and gene expression. Chromatin remodeling is partially regulated by histone modification, by various PTMs, which are as follows: acetylation, methylation, phosphorylation, PARylation, sumoylation, ubiquitylation and crotonylation, and chromatin remodeller complexes [39]. PARP1 is able to orchestrate these changes to chromatin architecture, and in turn chromatin-related functions. Below, we discuss the function of PARP1 in the regulation of histones (H1, core histone and histone variants) and chromatin remodeling enzyme complexes.

### 3.1. Linker Histone H1

The first association of PARylation with chromatin was reported as early as the 1960s [40,41]. In the early 1980s, Poirier et al. reported that PARP1 could PARylate H1 to promote the decondensation of native polynucleosomes in vitro, mimicking the effects of linker histone H1 depletion [42]. H1 has been identified as a major acceptor of PARylation in native chromatin [43]. In *Drosophila*, activated PARP1 strips chromatin proteins off DNA to form a puff structure, which makes transcription machinery complexes accessible to promoters [44]. However, H1 and PARP1 bind in a competitive manner to target gene promoters and have distinct roles in determining gene expression outcomes in vivo [45,46]. It is plausible that the highly negatively charged PAR on H1 induces the repulsion of nucleosomes, thereby leading to chromatin relaxation (Figure 2). For example, PARP1 cooperates with GATA3 to compete with linker histone H1 to maintain a transcriptionally competent chromatin environment for CCND1 [47]. H1 is PARylated and released at promoters of genes that are involved in the reprogramming of neuronal gene expression, which is required for learning and memory in the hippocampus of mice [48]. Hormone-dependent phosphorylation of PARP1 by CDK2 activates PARP1 to PARylate H1, which induces H1 release from chromatin. The opened chromatin is essential for the majority of progesterone-responsive genes [49]. Upon differentiation of mouse neuroprogenitor cells into neurons, PARP1 is recruited to the *doublecortin* (*Dcx*) promoter and mediates H1 PARylation and eviction of H1 from chromatin, thereby facilitating *Dcx* gene expression [25]. In primary mouse cortical neurons, KCl-induced depolarization results in H1 PARylation by PARP1, contributing to H1 release from chromatin and the simultaneous activation of immediate early gene (IEG) expression [50]. Activated PARP1 conducts PARylation and displacement of H1 from chromatin to form an euchromatin environment, leading to the activation of aromatase promoter I.3/II [51]. By using spFRET microscopy, PARP1 is found to interact with linker DNA of nucleosomes, promoting reorganization of the nucleosome, which is independent of PARylation activity [52]. Nucleosome reorganization occurs by PARP1 binding to linker DNA, where it displaces linker histone H1 to promote chromatin conformation that is permissive to gene expression [52]. Interestingly, PARP1 helps to maintain chromatin condensation, whereas linker DNA bound to PARP1 prefers to displace H1 to decondense chromatin at transcriptional active regions to facilitate transcription [53] (Figure 2).

### 3.2. Core Histones and Other Histone Variants

In addition to linker histone H1, PARP1 has been shown to PARylate all four core histones [43]. Early studies that focused on the PARylation site of core histones proposed that PARP1 modifies the important regulatory lysine residue of the core histone tails [54,55]. In LPS-stimulated macrophages, PARP1 enzymatic activity promotes gene transcription by increasing the promoter accessibility via PARylation of all four histones [56]. A recent study showed that PARylation of H2B-Glu35 by PARP1 inhibits AMPK-mediated phosphorylation of H2B-Ser36, which regulates proadipogenic gene expression and fat metabolism in vivo [57]. By PARylating H2B that hinders the occupancy of H2B at the *NFATc1* promoter, PARP1 represses the expression of the nuclear factor of activated T cell cytoplasmic 1 (NFATc1) and osteoclast differentiation [58].

Of note, the relationship between PARP1 and histones is complex. In addition to PARP1, which modifies histones by PARylation, histones and their PTMs can modulate the behaviors and activity of PARP1. In *Drosophila,* histones H2A and H2B inhibit the catalytic activity of PARP1, whereas H3 and H4 can bind to the catalytic domain of PARP1 to induce a H4-mediated induction of PARylation. H4 interacts with the C-terminal domain of PARP1, resulting in long-term activation of PARP1, and hence a sustained accumulation of PAR, which prolongs chromatin relaxation to facilitate the transcription factors’ access to DNA [59]. In *Drosophila*, the H2Av histone variant (*Drosophila* homolog of the mammalian H2AZ and H2AX variants) colocalizes together with *Drosophila* PARP1 (dPARP) on chromatin. Irradiation-induced phosphorylation of H2Av stimulates PARP1 enzymatic activity, which in turn triggers transcriptional activation [60]. Phosphorylation of H2Av by JIL-1 kinase increases the interaction between PARP1 and H4, leading to transcriptional initiation [26]. Moreover, the authors showed that, since H2A inhibits PARP1 activity by its *N*-terminal tail, the acetylation of H2A in the nucleosome disrupts the inhibitory effect of H2A on PARP1, thus promoting PARP1 activity [26]. Because of the high conservation of these Drosophila genes with mammals and humans, these results can be referenced for the function of PARP1 in chromatin remodeling and transcription in higher organisms. The non-histone domain (NHD) of macroH2A1.2 binds and inhibits PARP1 enzymatic activity, which silences an X chromosome [61]. Finally, PARP1 PARylates the lysine demethylase 5B (KDM5) and inhibits the binding of KDM5 to chromatin. This refracts the demethylation of H3K4me3 and maintains an open chromatin structure to positively regulate gene expression [62].

PARP1 has also a functional association with histone variants such as H2A.Z and macroH2A. The activation of the ERK pathway induces PARP1 activation and promotes binding of PARP1 to the *c-fos* promoter region, where PARP1 orchestrates the exchange of the histone variant H2A.Z with H2A to allow transcriptional activation in Hela cells and mouse embryonic fibroblasts (MEFs) [63]. In IMR90 primary human fetal lung fibroblast cells, PARP1 cooperates with macroH2A1.1 to regulate the transcription of macroH2A1-target genes, including *HDAC9*, *RGS4*, *CPA4*, *ANKRD1*, *EREG*, *FBLN1*, *FMO2* and *SHISA3*, by enhancing CREB-binding protein (CBP)/p300-mediated H2B acetylation. [64].

## 4. Chromatin Remodeling Complexes

PARP1 can function as a histone chaperone by recruiting chromatin remodeling enzymes, for example, histone PARylation factor 1 (HPF1 or C4orf27) and others that contain PAR-binding domains, to facilitate chromatin assembly upon DNA damage [65,66]. HPF1 is a key regulator of PARP1-dependent PARylation signaling, acting to promote serine ADP-ribosylation of histones, to limit DNA damage-induced PARP1 hyper-automodification, and to switch the protein modification from PARylation to MARylation [66,67,68,69]. HPF1-mediated serine ADP-ribosylation is a key step in DDR [21]. Of note, the blockage of PARP1 activity that inhibits the repair of DNA damages in tumor cells is a promising approach for cancer treatment [70,71]. Thus, it is worth studying inhibitors that target HPF1-dependent serine ADP-ribosylation. To date, there have been only studies of HPF1 regulating PARP1 in the context of DDR. Whether HPF1 is directly involved in PARP1-dependent transcription activation (see below) requires further investigation. PARylation can also recruit different chromatin remodeling enzymes via their PAR-binding domains [5,8], a range of motifs, including macrodomains, PAR-binding zinc finger (PBZ), PAR-binding motif (PBM), lysine/arginine (KR)-rich domain, high-mobility group (HMG) box-like domain, and PAR-binding regulatory motif (PbR) [5,72]. The macrodomain of chromatin-bound macroH2A1.1 binds to PAR that is locally produced by PARP1, resulting in macroH2A1.1-dependent chromatin compaction [73]. A recent study reported that macroH2A1.1 inhibits PARP1 activity, thereby preventing NAD^+^ depletion mediated necrosis, and meanwhile stabilizes PAR chains to facilitate DNA repair [74]. It is worth noting that unlike macroH2A1.1, macroH2A1.2 does not bind to PAR [75]. PAR binding by macroH2A1.1 represses cellular proliferation and regulation of gene expression [74,76]. MacroH2A1.1 is specially lost in a majority of cancer types, and thus acts as a tumor suppressor [76]. The interaction of macroH2A1.1 with PAR may explain its different function from its variant macroH2A1.2.

Chromodomain-helicase-DNA-binding protein (CHD) 1-like (CHD1L, also known as ALC1) is an SNF2-like ATPase that is recruited during DNA repair by PAR through its macrodomain motif to trigger chromatin relaxation [77,78]. Aprataxin- and PNK-like factor (APLF or C2orf13) is a well-known histone chaperon, which is recruited to DNA damage sites via interaction of its PBZ domains with PAR. It helps PARP1 to initiate a recruitment cascade of chromatin remodelers by facilitating ALC1 binding to histones and the recruitment of macroH2A1.1 to PAR [79]. CHD6 is another chromatin remodeler that is recruited to DNA damage sites via a conserved KR-rich domain that binds to PAR [80]. The human oncoprotein DEK is a non-histone chromatin architectural protein and, by its interaction with PAR via its PBM domain, it maintains a heterochromatic status in Drosophila [81]. CHD4 is a component of the nucleosome remodeling and deacetylase (NuRD) complex that binds to PAR with its high-mobility group (HMG) box-like domain and promotes the deacetylation of histone, in order to control chromatin reorganization and transcriptional repression [82,83].

As discussed above, PARylation generates long or short, branched or linear PAR, which recruits the chromatin remodeling factors to perform their functions. On the one hand, several chromatin remodeling complexes trigger chromatin relaxation, and thereby regulate the accessibility of DNA in favor of DNA repair. On the other hand, repressive chromatin modifiers are also recruited to sites of DNA damage, which by binding to PAR inhibits transcription. Therefore, the interaction spectrum between PAR and target proteins (PAR readers) may explain different functions of chromatin remodelers.

## 5. PARP1 as a Modulator of DNA Methylation

Mammalian DNA methylation is an epigenetic mechanism that involves the following two antagonizing processes: the transfer of 5-methylcytosine (5mC) on CpG dinucleotides (CG) by the DNA methyltransferase (DNMT) enzymes (DNMT1, DNMT3a, DNMT3b) [84,85], and demethylation by the action of the TET family of dioxygenases (TET1, TET2 and TET3). DNA methylation controls gene expression by altering chromatin conformation [86]. PARP1-mediated PARylation has been reported to coordinate the dynamics of either methylation or demethylation.

In 1997, Caiafa’s group provided the first evidence that PARylation is associated with DNA methylation [87]. A block of PARylation by PARP inhibitors induces hypermethylation of the CpG island in L929 mouse fibroblasts, while an active PARylation maintains the unmethylated state of the CpG island [88,89]. PAR from auto-PARylated PARP1 binds to DNMT1 and inhibits its enzymatic activity, which prevents DNMT1’s access to DNA, abrogating the methylation of CpG [90]. Chromatin insulator protein CTCF (CCCTC-binding factor) is a key factor in the functional interplay between PARP1 and DNA methylation [91]. CTCF stimulates PARylation activity of PARP1, which in turn maintains the unmethylated status of specific CTCF-bound CpGs, while inhibiting DNMT1’s activity [90,92]. Thus, PARP1 controls DNA methylation patterns by a combined regulatory mode of DNMT1 expression and activity (Figure 3A).

PARP1 is also involved in the maintenance of the unmethylated state of the regulatory sequences of other genes (Figure 3B,C). TET converts 5mC to 5-hydroxymethylcytosine (5hmC) that drives DNA demethylation [93]. Impaired PARP1 activity causes a significant reduction in the expression of TET1 [94]. In HEK293T cells, PARP1 is activated by interacting with TET1 independent of DNA damage; non-covalent PAR binding of TET1 resulted in the negative regulation of TET1 activity, whereas covalent PARylation had a stimulatory effect on TET1 activity [95]. In PARP1 deleted cells, increased mRNA expression of *Tet* correlates with an increased level of 5hmC and *Cxcl12* promoter demethylation increases the *Cxcl12* expression [96]. Of note, the histone demethylase KDM5A binds non-covalently with PAR, presented by chromatin remodeler NuRD after PARylation in response to DNA damage, and promotes demethylation of H3K4me3, which stabilizes NuRD at chromatin [97].

The crosstalk between PARP1 activity and DNA demethylation is crucial for gene transcription during cellular differentiation. For example, the interaction of PARP1 with TET2 stimulates conversion of 5mC to 5hmC, which is crucial for early-stage epigenetic modification during somatic cell reprogramming [98]. NAD^+^ supplementation stimulates PARP1 activity and produced PAR that inhibits DNMT1 activity, resulting in demethylation and transcriptional activation of the *CEBPA* gene that may be related to myeloid differentiation [27]. Hyperactivation of PARP1 is associated with impaired DNA methylation processes in peripheral blood mononuclear cells from type 2 diabetes mellitus patients [99]. PARylation regulates methylation of histone H3 and DNA methylation during the first cell cycle of mouse development [100]. The mutual exclusivity between the binding of PARP1 to chromatin genome-wide and DNA methylation pattern suggests a functional interplay between PARP1 and DNA methylation [101].

## 6. PARP1 and PARylation Regulate Gene Transcription

In addition to its involvement in the processes of DNA repair and chromatin homeostasis, PARP1 has been shown to play a prominent role in gene transcription by interacting with and modulating transcriptional machineries at gene promoters. The PARP1 protein directly interacts with numerous transcription factors, such as NF-κB [102], HES1 [103], Elk-1 [24], SOX2 [104], NFAT [105], and AP-1 [106]. PARP1 can regulate transcription positively with co-activators or negatively with repressors. Thus, PARP1 regulates many signaling pathways, thereby controlling a wide array of patho-physiological processes [107].

### 6.1. PARP1 in Inflammatory Response

For a long time, PARP1 has been shown to be a key mediator of inflammatory response [108,109]. In response to LPS treatment, PARP1^−/−^ mice were resistant to septic shock because of the greatly reduced expression of pro-inflammatory cytokines TNF, IL-6 and iNOS [110]. PARP1 also promotes the pathological inflammatory response in the central nervous and cardiovascular systems [111]. *Drosophila* that lacks PARP showed insufficient innate immune response and were under an increased risk of bacterial infection [44].

The Hottiger lab first described the mechanism of PARP1 in inflammatory response and demonstrated that PARP1 acts as a transcriptional co-activator to bind both p50 and p65 subunits of NF-κB [112]. PARP1 also interacts with histone acetyltransferases CBP/p300 to promote PARP1’s interactions with p50, which results in NF-κB activation [113,114]. Not only via its protein scaffold, PARP1’s enzymatic activity also influences NF-κB-dependent transcription. However, the conclusion of PARP1 or PARylation activity in NF-κB signaling is complex. An early report using cell extracts showed that PARP1 repressed the DNA-binding activity of NF-κB and PARylation of p50 and p65 inhibited NF-κB-dependent transcription [115]. Many other studies show that PARP1 is positively involved in the NF-κB pathway. For example, in murine macrophage cells, after LPS treatment, PARP1 PARylates p65 and increases the NF-κB-mediated transcription of inflammatory cytokines (such as Il-1β and Il-18) [116]. PARP1 auto-PARylation enhances the DNA-binding activity of p50 [117]. In mouse glia cells, auto-PARylated PARP1 and PARP1 activity increase the LPS-induced DNA binding of NF-κB and the production of cytokines TNFα and iNOS [118]. PARP1 knockout and knockdown, as well as PARP1 inhibitors, can all blunt the interaction between p65-NF-κB and exportin Crm1, thereby attenuating p65 NF-κB nuclear retention, ultimately reducing the expression of NF-κB -targeted genes, such as iNOS and ICAM-1, in LPS-stimulated mouse smooth muscle cells [119]. In the lumbar 5 spinal nerve ligation (SNL) model, SNL increases histone H1 PARylation and PARP inhibitors or PARP1 knockdown repress the binding of NF-κB p65 at the *TNFα* promoter and downregulates TNFα expression in dorsal root ganglia and spinal dorsal horn [120]. Furthermore, LPS treatment induces PARP1-mediated PARylation of histones at transcription active chromatin regions and the promoters of Il-1β, MIP-2 and CSF2, which facilitates NF-κB recruitment to these promoters [56]. Interestingly, a caspase-resistant PARP1 impairs NF-κB-mediated transcription activity and proinflammatory cytokine production in response to LPS-induced sepsis and intestinal and renal ischemia-reperfusions [121], indicating that PARP1 cleavage at the DNA-binding domain is involved in NF-κB transcription activation.

Apart from auto-PARylation, phosphorylateion of PARP1 by c-Abl under inflammatory agent exposure activated PARylation of p65 and the transcription of NF-κB-target genes [122]. In addition, PARP1 activation achieved by ERK2-mediated phosphorylation, in turn, enhances ERK-induced Elk1-phosphorylation and the transcription of the Elk1-target gene *c-fos* [24]. Similar to the LPS-mediated sepsis model, PARP1^−/−^ mice were protected from ulcerative colitis induced by trinitrobenzene sulfonic acid (TNBS) treatment, because PARP1 can activate c-Jun of the AP-1 transcription factor [123]. PARP1 can also interact with and PARylate the transcription factor NFAT and increase the binding of NFAT to DNA. PARP1 deficiency or PARP1 inhibitors compromise the expression of cytokines [105,124]. However, PARP1-mediated PARylation of STAT3 inhibits the transcriptional activity of STAT3 and suppresses the expression of PD-L1 [125,126]. The scaffold function of the PARP1 protein and its enzymatic activity seem to complicate the role of PARP1 in transcriptional activation. Nevertheless, these studies demonstrate multiple layers of the gene regulation scheme by PARP1 and PARylation.

### 6.2. Embryonic Development and Cell Differentiation

Despite the normal development of PARP1^−/−^ mice, PARP1/PARP2 double knockout mice that mostly lack PARylation capacity were embryonic lethal [127], indicating that PARylation homeostasis is crucial for embryo development. The specific tempo-spatial mode of transcription is important for embryonic development. Transcription factors play an important role in maintaining the pluripotency of embryonic stem cells. PARP1 interaction or PARylation modulates the affinity of transcription factors to DNA, and thereby their transcription activity. PARP1 interacts with and PARylates SOX2, thereby reducing SOX2 binding at the enhancer of *FGF4* to promote the expression of *FGF4* [104]. Another study, however, showed that PARP1 cooperates with SOX2 to facilitate its binding to poorly accessible chromatin and maintains pluripotency in mouse embryonic stem cells, independent of its catalytic activity [128]. The transcription regulation of specific genes dictates cell differentiation. PARP1-mediated PARylation dissociates the SMAD complex from DNA and impairs TGFβ in the differentiation and migration of cells during embryonic development [129]. An activation of calcium-dependent protein kinase (CaMKIIδ) induces phosphorylation and the activation of PARP1, which dissociates the PARylated TLE1 corepressor from HES1 to switch the HES1 function from gene repression to gene activation during neuronal differentiation [103]. In addition, PARP1 directly interacts with ERK2, which then enhances phosphorylation of the transcription activator ELK1 to activate *c-fos* in growth factor stimulated neural cells and cardiomocytes [24]. PARP1 PARylates C/EBPβ, a key pro-adipogenic transcription factor, and abrogates its DNA-binding and transcriptional activities, thereby affecting the differentiation of adipocyte precursors [130].

### 6.3. Other Cellular Processes

PARP1 or PARylation also participate in the transcription regulation of many other genes and cellular processes, such as in fat production, cardiovascular and cancer formation. The binding of PARP1 to transcription factor E2F-1 promotes the expression of MYC (c-Myc) for tumorigenesis [131]. PARP1 binds and PARylates FOXO1 to inhibit FOXO1-induced transcription of cell cycle inhibitor p27^Kip1^ [132]. PARP1 can PARylate estrogen receptor alpha (ERα) and promote its binding to the estrogen reaction element (ERE) in the promoter to activate the transcription of ERα-mediated genes for cardiovascular protection [133]. PARP1 binds to MAF proteins and the antioxidant response elements (ARE) of Nrf2 target genes to activate the transcription of Nrf2 target genes [134]. PARP1, as a co-repressor of transcription together with MATIIα, represses the expression of MAF and Bach1 target genes [135]. PARP1 interacts with the *mEH* gene (*EPHX1*) proximal promoter and the linker histone complex H1.2 to stimulate *EPHX1* transcription [136]. PARP inhibitor (olaparib, rucaparib) administration decreased the activity and expression of the stress sensor LKB1, although the underlying mechanisms are unknown [137].

In summary, PARP1 can modulate gene expression by direct binding and/or PARylating transcription factors to control gene transcription in response to extracellular signals. This can be achieved by modulating transcription factors’ stability, PTM, or chromatin conformation, all of which affect the DNA binding and/or activity of these transcription machineries.

## 7. Perspectives

Over the past 60 years, the biochemical and biological functions of PARP1 and PARylation have been extensively studied. The most well-known function is evidently its DNA damage detection, repair and signaling. This function is prominent because PARP1 is largely activated by acute DNA damage and PAR formation. Because of PARP1’s auto-PARylation activity in the process of the DNA damage response and repair, PARP inhibitors have been used to develop a “synthetic lethality” strategy to treat cancers that carry genetically mutated DSB repair pathways, e.g., BRCAs [71,138,139]. Several PARP inhibitors have been recently approved by the Food and Drug Administration (FDA) and European Union (EU) for their efficacy against a variety of cancers [70,71]. However, a growing amount of studies has revealed an important role of PARP1 in other biochemical activities, such as chromatin remodeling, DNA methylation and transcriptional regulation, independent of DNA damage. PARP1 is an abundant and stable nuclear protein. PARP1 changes the chromatin architecture by modifying histones and influencing DNA methylation to facilitate the compaction or loosening of the chromatin structure. PARP1 participates in transcription either as a co-activator or co-repressor, depending on the DNA-binding sequence specificity of the transcription factors, histones of promoter regions, and its interaction with the subunits of the transcription machinery. In addition, the dynamic production and degradation of PAR polymers can influence PAR-binding proteins to specific chromatin loci during transcription. Moreover, PARP1 PARylates target proteins to regulate their activity at these chromatin sites or the promoters. For the same protein that interacts with PARP1, covalently or non-covalently binding of PAR may result in a different outcome of transcription activation. We need a better characterization of the specific targets of PARylation at the promoters in a chromatin context to understand how PARP1 and PARylation change the behaviors of chromatin conformation and transcription factors, under diverse physiological and pathological conditions.

## 8. Conclusions

PARP1 is well documented to be activated not just by damaged DNA but also by various environmental and developmental stimuli. PARP1 has been broadly related to gene expression, for example by PARylating histones and chromatin remodeling proteins that modulate chromatin architecture and behaviors of transcription machineries at gene promoters. PARP1 controls expression of many genes operating immune responses, cell survival and inflammation. The function of PARP1 in gene regulation thus may contribute to the establishment of the molecular actions of PARP inhibitors for their clinical applications in treatment of inflammatory diseases, as well as, cancer.

## Figures and Tables

**Figure 1 cancers-14-04162-f001:**
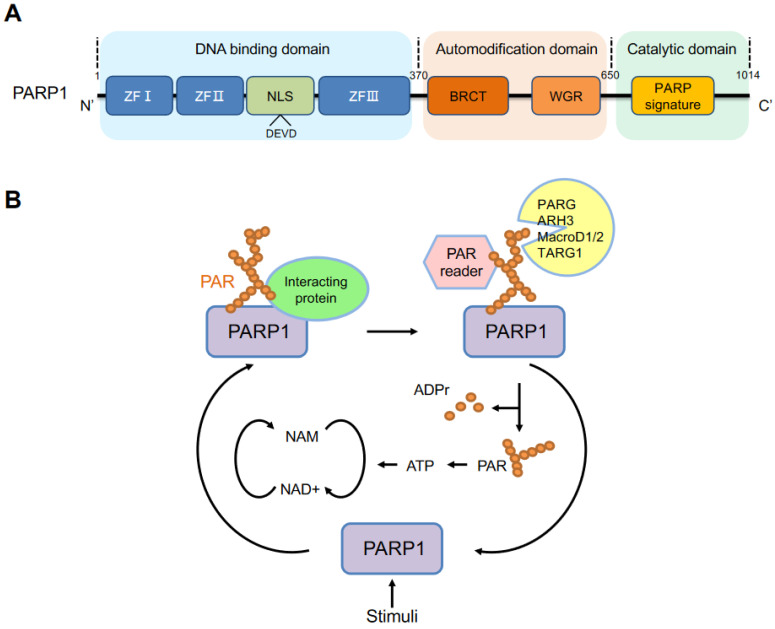
PARP1 structure and PARylation. (**A**) Structural and functional domains of human PARP1. ZFI, ZFII and ZFIII: zinc finger motifs I, II and III, respectively; NLS: nuclear localization signal; DEVD: a caspase cleavage site; BRCA1 C-terminus (BRCT); WGR (Trp–Gly–Arg). A highly conserved PARP signature is located in the C-terminal catalytic domain. (**B**) The cycle of PARylation as depicted by auto-PARylation of PARP1, a well-documented acceptor. Stimuli, including but not limit to, DNA damage, ERK1/2, hormone, JIL-1, activate PARP1, which catalyzes the formation of long and branched poly (ADP-ribose) (PAR), using NAD^+^ as a substrate. PARP1 can interact with other partners, either by itself as a scaffold protein or via its enzymatic product PAR. PAR can be recognized by PAR readers and be rapidly degraded by hydrolytic enzymes, PARG, ARH3, MacroD1 and D2, TARG1. NAM: nicotinamide; ADPr: ADP-ribose; PAR: poly (ADP-ribose).

**Figure 2 cancers-14-04162-f002:**
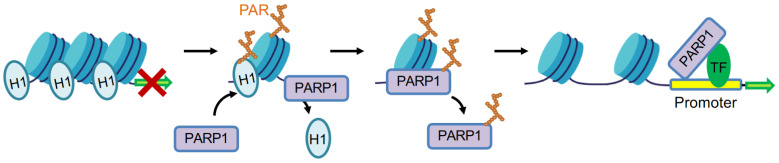
PARP1 regulates transcription by facilitating local chromatin decondensation at active gene sites. Activated PARP1 mediates H1 PARylation and displacement from chromatin to form euchromatin environment. PARP1 also physically interacts with transcription factors (TF), with or without its enzymatic activity, to modulate gene transcription.

**Figure 3 cancers-14-04162-f003:**
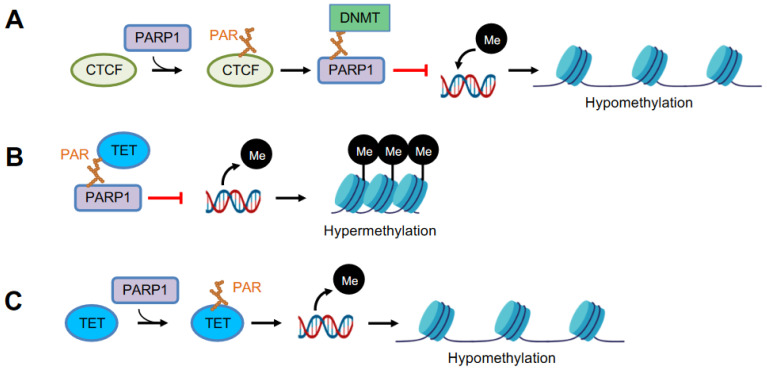
PARP1 modulates DNA methylation. (**A**) PARP1 PARylates chromatin insulator CTCF (CCCTC-binding factor), which in turn stimulates auto-PARylation of PARP1. Auto-PARylated PARP1 inhibits the catalytic activity of DNMT through non-covalent binding between PAR-DNMT, resulting in DNA hypomethylation. (**B**) Non-covalent binding of TET and PAR negatively regulates TET activity, resulting in DNA hypermethylation. (**C**) Covalent PARylation of TET promotes TET1 activity, causing DNA hypomethylation.

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
