# Peer review of "PARP1: Liaison of Chromatin Remodeling and Transcription"

_cancers, 2022, doi:10.3390/cancers14174162_

Round 1
Reviewer 1 Report
The authors have comprehensively described how PARP1 and polyADP-ribosylation are involved in gene regulation and chromatin remodeling. In the first part, the general concept of chromatin remodeling with the state-of-art examples are recommended to be added for the better understanding.
-Page 2: The ref.12, which described only partial cDNA cloning, should be replaced by the reference of human PARP1 cDNA cloning paper (PMID2824474, Kurosaki et al., 1987).
Page 4 ref
- Page 4: For ref 22 and 23, original articles should be added as the references from the current list.
- Page 4: For PARG, ARH3, MacroD1, and MacroD2, original articles are recommended to be cited for these important enzymes.
- Page 4, line 15, the description for ARH3 function should be also added.
Fig. 2A and B: The direction of bent arrows for “Me“ seems to be opposite and should be corrected.
- Page 5: “PTMs” should be defined.
- Page 5, section title of 3: The term “chromatin remolding” should be corrected to “chromatin remodeling”.
-Typos including followings should be corrected:
Page 5: histon
Page 5: histones modifications → histone modification
Page 5: IEG needs explanation of abbreviation
Page 7: DNA damaged → DNA damage
Page 10: Olaparib, Rucaparib → olaparib, rucaparib
Page 6, Section title of 6 : regulates → regulate
Reviewer 2 Report
In this manuscript, the authors summarize the biological functions of PARP1 in chromatin remodeling and transcriptional regulation. Poly(ADP-ribosyl)ation is a unique post-translational modification that is catalyzed by PARP family enzymes, especially the funding member PARP1. Here, the authors comprehensively introduce the structure of PARP1 as well as the molecular mechanism of PARP1 activation. Moreover, the authors summarize the major chromatin substrates of PARP1, namely linker histones and nucleosomal histones. The authors also include the evidence of the crosstalk between poly(ADP-ribosyl)ation and other signaling pathways, such as DNA methylation/demethylation in the context of chromatin biology and transcription. Overall, this is a well written manuscript that include many lines of evidence to demonstrate the role of PARP1 on chromatin remodeling and gene transcription. Some suggestions are listed below:
1. It would be better to include a citation on how free PAR is recycled for ATP production (bottom line of Page 4).
2. A title of Figure 1 may be needed.
Reviewer 3 Report
This review article by Wen Zong and colleagues discusses recent advances in the chromatin remodeling and transcription functions of PARP1, particularly those which are independent from its role during the DNA damage response. This is an exciting topic given the rapid emergence of PARP1 functions outside of the DDR, including the regulation of disease associated genes. This review covers roles for PARP1 in chromatin structure/modification, DNA modifications (with focus on methylation) and transcriptional regulation during inflammation and development. While this review is well-written and clearly highlights many important findings being timely in subject matter, the significance of the results presented could be better discussed. The potential impact and interest from the field for this review would be improved by more consideration of the context and significance of the cited work to make a more balanced and thorough overview of this important topic.
Specific Comments
Major
1. The review would benefit from more summary/discussion of significance at the end of each section or paragraph, many relevant studies are cited but insight into how they impact overall understanding is lacking.
2. NuRD also requires PARylation for recruitment to DNA damage sites (PMID:20937877). This includes through the ability of KDM5A to bind PAR chains through a newly identified PAR binding Coiled-coil domain (PMID:34003252). This has been shown to lead to demethylation of H3K4me3 to allow stable association of NuRD to DNA damage sites as previous work showed that H3K4me3 repulses NuRD binding onto chromatin. This pathway also is likely to be involved in gene regulation and has been reviewed recently (PMID: 35532219). These citations should be included in the review.
3. Section 3.2 is highly focused on drosophila studies, chromatin structure and histone modifications in drosophila can differ significantly from human chromatin and the relevance of these findings to humans should be discussed more in depth.
4. Section 4 on chromatin remodeling provides very little insight into recent studies despite this being in the title. The PARP1 accessory factor HPF1 is mentioned but recent studies (including: https://www.nature.com/articles/s41467-021-26172-4) linking this factor to transcriptional regulation outside of DDR should be discussed.
5. Several PAR binding domains are listed in section 4 however the functional significance of these binding events in chromatin remodeling or transcription is not discussed.
6. For DNA repair pathways, Alternative end-joining (Alt-EJ) should also be mentioned as it is an important sub-pathway of DSB repair that is dependent on PARP1.
7. It would be useful to mention that the splice variant of macroH2A, macroH2A1.2 does not bind PAR unlike macroH2A1.1. This provides an interesting regulatory mechanism to impact macroH2A splice variants through differential PAR binding abilities.
8. As DNMT1 methylates DNA during DNA replication, it would be useful to put these studies into this context. The majority of PARylation in S-phase occurs due to Okazaki fragment maturation (PMID:299833212). Is there a connection here?
9. It seems a missed opportunity to not introduce and discuss PARPi that are used clinically. This is one of the most important aspects of PARP1 currently and should be included. The discussion of PARP1 regulation of transcription in cancer should be expanded. The article concludes with speculation regarding the use of PARP inhibitors for cancer treatment, but the relevant literature and mechanisms are not discussed throughout the article.
Minor
1. In figure 1 the authors could also show PARylation onto various substrates and not just auto-PARylation in fig 1B.
a. Auto-PARylation on Serine residues should be included given that Serines have been proposed to be the major site of PARylation during DNA damage activation.
2. Section 3.1: “It is plausible that the highly negatively charged PAR onto H1 induces repulsion of DNA”
3. In the discussion of reference 51 the author’s state: “the acetylation of H2A in the nucleosome disrupts the inhibitory effect of H2A on PARP1…” without introducing this relationship between PARP1 and H2A first. This makes the significance of this finding difficult to interpret.
4. Title for section 4 should be: Chromatin remodeling complexes
5. In general, the figures appear to be low resolution
6. Page 5, 2nd sentence histon – histone
7. For PTMs, Ubiquitylation should be included.
8. Figure 3 – DMNT should be DNMT
Round 2
Reviewer 3 Report
The authors have now addressed all of our previous concerns. This review is now appropriate for publication.